# Selection of Reference Genes in *Evodia rutaecarpa* var. *officinalis* and Expression Patterns of Genes Involved in Its Limonin Biosynthesis

**DOI:** 10.3390/plants12183197

**Published:** 2023-09-07

**Authors:** Yu Zhou, Yuxiang Zhang, Detian Mu, Ying Lu, Wenqiang Chen, Yao Zhang, Ruiying Zhang, Ya Qin, Jianhua Yuan, Limei Pan, Qi Tang

**Affiliations:** 1National Research Center of Engineering Technology for Utilization of Botanical Functional Ingredients, College of Horticulture, Hunan Agricultural University, Changsha 410128, China; m18574824089@163.com (Y.Z.); m13787856952@163.com (Y.Z.); mudetian12580@163.com (D.M.); luying960522@163.com (Y.L.); 15958699677@163.com (W.C.); ningzby@163.com (Y.Z.); zry980610@163.com (R.Z.); 2Guangxi Botanical Garden of Medicinal Plants, Nanning 530023, China; qygxyyzwy@163.com; 3Changsha Hemao Agricultural Development Co., Ltd., Ningxiang County, Changsha 410609, China; 19507495178@163.com

**Keywords:** *E. rutaecarpa* var. *officinalis*, limonin, biosynthesis pathway, reference genes, expression patterns, qRT-PCR

## Abstract

*E. rutaecarpa* var. *officinalis* is a traditional Chinese medicinal plant known for its therapeutic effects, which encompass the promotion of digestion, the dispelling of cold, the alleviation of pain, and the exhibition of anti-inflammatory and antibacterial properties. The principal active component of this plant, limonin, is a potent triterpene compound with notable pharmacological activities. Despite its significance, the complete biosynthesis pathway of limonin in *E. rutaecarpa* var. *officinalis* remains incompletely understood, and the underlying molecular mechanisms remain unexplored. The main purpose of this study was to screen the reference genes suitable for expression analysis in *E. rutaecarpa* var. *officinalis*, calculate the expression patterns of the genes in the limonin biosynthesis pathway, and identify the relevant enzyme genes related to limonin biosynthesis. The reference genes play a pivotal role in establishing reliable reference standards for normalizing the gene expression data, thereby ensuring precision and credibility in the biological research outcomes. In order to identify the optimal reference genes and gene expression patterns across the diverse tissues (e.g., roots, stems, leaves, and flower buds) and developmental stages (i.e., 17 July, 24 August, 1 September, and 24 October) of *E. rutaecarpa* var. *officinalis*, LC-MS was used to analyze the limonin contents in distinct tissue samples and developmental stages, and qRT-PCR technology was employed to investigate the expression patterns of the ten reference genes and eighteen genes involved in limonin biosynthesis. Utilizing a comprehensive analysis that integrated three software tools (GeNorm ver. 3.5, NormFinder ver. 0.953 and BestKeeper ver. 1.0) and Delta Ct method alongside the RefFinder website, the best reference genes were selected. Through the research, we determined that *Act1* and *UBQ* served as the preferred reference genes for normalizing gene expression during various fruit developmental stages, while *Act1* and *His3* were optimal for different tissues. Using *Act1* and *UBQ* as the reference genes, and based on the different fruit developmental stages, qRT-PCR analysis was performed on the pathway genes selected from the “full-length transcriptome + expression profile + metabolome” data in the limonin biosynthesis pathway of *E. rutaecarpa* var. *officinalis*. The findings indicated that there were consistent expression patterns of *HMGCR*, *SQE*, and *CYP450* with fluctuations in the limonin contents, suggesting their potential involvement in the limonin biosynthesis of *E. rutaecarpa* var. *officinalis*. This study lays the foundation for further research on the metabolic pathway of limonin in *E. rutaecarpa* var. *officinalis* and provides reliable reference genes for other researchers to use for conducting expression analyses.

## 1. Introduction

*Wuzhuyu*, as one of the traditional Chinese medicines, refers to the dry and nearly mature fruit of *Evodia rutaecarpa* (Juss.) Benth, *Evodia rutaecarpa* (Juss.) Benth. var. *officinalis* (Dode) Huang, or *Evodia rutaecarpa* (Juss.) Benth. var. *bodinieri* (Dode) Huang. It mainly contains alkaloids, volatile oil, flavonoids, and limonoids and was originally used to treat headaches, amenorrhea, postpartum bleeding [1], and gastrointestinal diseases [2,3]. The latest research shows that it also has therapeutic effects on dysmenorrhea [4] and Alzheimer’s disease [5]. According to the Chinese Pharmacopoeia, the contents of evodiamine and rutaecarpin in the dry products of *E. rutaecarpa* var. *officinalis* must not be less than 0.15% and those of limonin must not be less than 0.20% [6]. *E. rutaecarpa* var. *officinalis* is mainly distributed in the south of China. Most of the research on *E. rutaecarpa* var. *officinalis* has focused on the extraction, separation [7], and identification of its chemical components [8] and the pharmacology [9] and toxicology of its effective components [10]. Limonin, found in *E. rutaecarpa* var. *officinalis*, exhibits anti-inflammatory, antibacterial, and antioxidant properties [11] and has a wide range of effective anti-cancer effects [12]. The method for improving the limonin content in *E. rutaecarpa* var. *officinalis* through traditional cultivation and breeding is inefficient and costly. The genes involved in the biosynthesis of limonin in *E. rutaecarpa* var. *officinalis* have not yet been investigated. Analyzing its molecular mechanism represents an effective means to improve the content of limonin in *E. rutaecarpa* var. *officinalis*.

There have not yet been studies on the molecular mechanism of the biosynthesis of the active ingredients in *E. rutaecarpa* var. *officinalis*. The biosynthesis pathway of limonin has a long history of research, but no one has taken *E. rutaecarpa* var. *officinalis* as the research object. This study is the first to study limonin in *E. rutaecarpa* var. *officinalis*. It can provide a new objective and approach for many researchers to study limonin. Limonin is a tetracyclic triterpene compound. Terpenoids are compounds, and their derivatives are based on isoprene as the basic unit. Isopentenyl diphosphate (IPP) and dimethylallyl diphosphate (DMAPP) are two precursors of isoprenoids. These precursors can be synthesized via two different pathways: the mevalonate (MVA) pathway and the 2-C-methyl-D-erythritol 4-phosphate (MEP) pathway. The biosynthesis of limonin begins with the generation of IPP and DMAPP through both the MVA and MEP pathways. In the MVA pathway, the substrate acetyl CoA is successively catalyzed by ACAT, HMGCS, HMGCR, MVK, MVAK, and MVD to form IPP. In the MEP pathway, the substrate D-Glyceraldehyde-3-phosphate undergoes catalysis by DXS, DXR, ISPD, ISPE, ISPF, ISPG, and ISPH to form DMAPP. Both IPP and DMAPP are then catalyzed by geranylpyrophosphate synthetase (GPS) to produce geranylpyrophosphate (GPP). Subsequently, IPP and GPP react under the catalysis of farnesyl pyrophosphate synthetase (FPS) to form farnesyl pyrophosphate (FPP). Two molecules of FPP can combine to generate presqualene diphosphate under the catalysis of squalene synthase (SQS). Presqualene pyrophosphate is further catalyzed by squalene epoxidase (SQE) to form 2,3-oxidosqualene [13]. The pathways of limonoids have been elucidated, but limonin’s have not yet been elucidated. In *E. rutaecarpa* var. *officinalis*, limonin is obtained from intermediates such as melianol, nomilin, and obacunone. It is possible that CYP450 enzymes are involved in the intermediate steps of limonin synthesis from melianol [14]. The potential biosynthesis pathway of limonin in *E. rutaecarpa* var. *officinalis* is illustrated in Figure 1.

In order to study the enzyme genes related to the biosynthesis pathway of limonin in *E. rutaecarpa* var. *officinalis*, it is necessary to validate the functions of the candidate genes. The research on gene expression is a crucial approach for understanding gene functions. Conducting gene expression studies requires reliable reference genes. The optimal reference genes can vary and exhibit fluctuations across different species, among diverse tissues, across various anatomical regions, across different developmental stages, and under distinct treatments within the same species. It is necessary to screen for reference genes under different conditions. The stability of these reference genes must be analyzed. Real-time quantitative PCR (qRT-PCR) is commonly used for gene expression analysis due to its high accuracy, strong specificity, and simple operation [15,16,17]. With the application of qRT-PCR technology, the reference genes were identified in various plant species. For instance, in *Fragaria chiloensis*, *chloroplastic BYPASS regulatory factor (FcCHP) 2* and *FcACTIN1* were identified as the most stable reference genes in vegetative samples, while *FcDBP (DNA binding protein)* and *FcCHP1* were found to be stable in ripening fruit samples [18]. Similarly, in *Kengyilia melanthera*, *translationally controlled tumor protein (TCTP)* and *TIP41-like protein (TIPRL)* were the most stable reference genes under abscisic acid (ABA) treatment, *TIP41-like protein (CACS)* and *TCTP* were the most stable reference genes under cold treatment, *CACS* and *F-box only protein 6-like (FBXO6L)* were the most stable reference genes under both heat and drought treatment, and *TIPRL* and *F-box only protein 6-like (CYPA3)* were the most stable reference genes under salt treatment [19]. *F1-ATPase/ATP synthase/ACCase (Acetyl CoA Carboxylase)* were identified as the best reference genes in *S. rosmarinus* [20]. In *ginseng*, different vegetative organs have shown variations in the most stable reference genes. For the roots, *GAPDH* and *30S ribosomal protein S20 (30SRPS20)* were found to be the best reference genes; in the stems, *cyclophilin (CYP)* and *(60S ribosomal protein L13 (60SRPL13)* were top-ranked; and in the leaves, *CYP* and *ubiquinol-cytochrome C reductase (QCR)* exhibited the highest levels of stability [21].

In this study, the stability of ten candidate reference genes was evaluated, and the two most stable reference genes were used to relatively quantify the genes related to the limonin biosynthesis pathway that were screened according to the “full length transcriptome + expression profile + metabolome”, and the expression profiles of the genes related to the limonin biosynthesis pathway were obtained. Based on the changes in gene expression, it was inferred that several genes may be involved in the biosynthesis of limonin in *E. rutaecarpa* var. *officinalis.*

## 2. Results

### 2.1. Analysis of Relative Content of Limonin in E. rutaecarpa var. officinalis

The fruits of *E. rutaecarpa* var. *officinalis* from the four periods were categorized as early stage (17 July), middle stage (14 August), late stage (1 September), and final stage (24 October). The fruits gradually increased in size as they grew, transitioning from a tender green color in the early stage, progressing to yellow in the middle stage, and, finally, maturing into a reddish-brown hue. In terms of production, the optimal harvesting period for *E. rutaecarpa* var. *officinalis* fruits is generally from late August to early September. It is ideal to harvest the fruits during the color transition from green to yellow. As shown in Figure 2, limonin was not detected in the roots, stems, leaves, or buds. The limonin contents in the fruits decreased as the fruits matured. Additionally, the limonin contents in the fruits during the four periods reached an extremely significant level of difference (*p* < 0.01).

### 2.2. Selection of Reference Genes and Primer Design

The amplification efficiency of the ten reference gene primers ranged from 94.4% to 111.5%. The linear correlation coefficients were all greater than 0.98, and the dissociation curves each showed a single peak (see Appendix A). These results indicated that the primers exhibited good specificity and were suitable for use in the subsequent experiments. The primer sequence, correlation coefficients, and amplification efficiency of the qRT-PCR for the ten candidate reference genes are shown in Table 1.

### 2.3. Expression Abundance Analysis of the Candidate Reference Genes

To assess the expression abundance of the ten candidate reference genes, qRT-PCR was performed using ten-fold diluted cDNA extracted from the roots, stems, leaves, flower buds, and fruits at the different developmental stages as templates. The cycle threshold (Ct) values obtained directly reflected the expression levels of each candidate reference gene, and the expression abundance was inversely correlated with Ct values, while lower Ct values indicated higher gene expression levels. As shown in Figure 3, the average Ct value of *UBQ10* was 19.65, which was the lowest among all genes. All of its Ct values ranged from 18.18 to 22.91, highlighting *UBQ10* as the candidate gene with the highest expression abundance. Conversely, the average Ct value of *Act100* was 27.06, which was the highest among all the genes. All of its Ct values ranged from 22.37 to 30.93, indicating that *Act100* had the lowest expression abundance. The lengths of the box plots in Figure 3 represent the data distribution. A shorter box plot suggests that the data were concentrated within a narrow range while a longer box plot indicates more dispersed data with significant differences. Although *UBQ10* exhibited the lowest expression abundance, *Act1*, *His3*, and *UBQ* showed better stability.

### 2.4. Stability Analysis of the Candidate Reference Genes

#### 2.4.1. GeNorm Analysis

GeNorm ver. 3.5 [22] is a software tool utilized for screening reference genes and determining the optimal number of reference genes required for qRT-PCR experiments. This program can be employed to evaluate any number of reference genes within various experimental setups. Ultimately, it selects combinations of two or more reference genes to normalize the data, enhancing the accuracy of relative quantification outcomes. Prior to GeNorm analysis, 2^−ΔCt^ values are computed and then imported into the software. GeNorm identifies the reference genes with stable expression by calculating the M value for each reference gene [23]. The rule of thumb is that a lower M value indicates better stability while a higher M value suggests lower stability. An M value below 1.5 signifies stable gene expression. The software also calculates the paired variation (V) value after the inclusion of each new reference gene and employs the V_n_/V_n+1_ ratio to determine the optimal number of reference genes required. The default threshold is set at 0.15. If the value of V_n_/V_n+1_ is less than 0.15, then the optimal number of reference genes is ‘n’. If the V_n_/V_n+1_ ratio exceeds 0.15, then the optimal number of reference genes is ‘n+1’ [24]. If all combination ratios surpass 0.15, then this threshold can be adjusted based on the experimental conditions [25].

As shown in Figure 4, the M values of all the genes were below 1.5, indicating that the stability of these candidate reference genes. *UBQ*, *β-TUB6*, and *UBQ10* exhibited better stability across the different developmental periods compared to the different tissues, whereas *His3*, *Act100*, *CHS1*, and *α-TUB* demonstrated an opposite trend. The GeNorm analysis results indicated that *Act1* and *GAPDH* were the two most stable genes across both scenarios. This is depicted in Figure 5. For both the different developmental periods and the diverse tissues, the *V*_2/3_ ratios were less than 0.15, implying that the optimal number of reference genes for normalization was two.

#### 2.4.2. NormFinder Analysis

In the application of the NormFinder ver. 0.953 [26] program, similar to GeNorm, the calculation of 2^−ΔCt^ is a prerequisite. This calculation is illustrated in Figure 6. The results obtained from the NormFinder analysis revealed that the highest stability rankings among the fruits at the distinct developmental stages were attributed to *Act1*, *UBQ*, and *GAPDH*. Meanwhile, in various tissues, the top stability rankings were associated with *Act1*, *His3*, and *GAPDH*. Remarkably, Act1 consistently maintained its first position in both scenarios. Conversely, *CHS1* and *α-TUB* demonstrated poor stability in the fruits at the different developmental stages, rendering them unsuitable as reference genes. Among the ten candidate reference genes, *Act1* emerged as the most stable and could serve as a reliable normalization reference gene.

#### 2.4.3. BestKeeper Analysis

The software Bestkeeper ver. 1.0 [27] employs the Ct value of a potential reference gene to directly compute the standard deviation (SD) and coefficient of variation (CV). This process is presented in Table 2. Smaller CV ± SD values correspond to greater stability exhibited by a reference gene. Among the fruits at the distinct developmental stages, *UBQ* demonstrated optimal stability, whereas among the different tissue types, *His3* emerged as the most stable reference gene. Intriguingly, Act1, which secured the top rank in both the GeNorm and NormFinder analyses, claimed the second position in the BestKeeper analysis. Conversely, *UBQ* ranked highest across the different periods, showcasing inferior performances within the varying tissues.

#### 2.4.4. Delta Ct Analysis

The Delta Ct analysis method involves utilizing the Ct value of a potential candidate gene to compute the average standard deviation for its stability ranking. A lower average standard deviation indicates greater gene stability. As shown in Figure 7, within the pool of the ten candidate reference genes, *Act1* emerged as the most stable, whereas *α-TUB* exhibited the least stability among the fruits across the various developmental periods.

#### 2.4.5. RefFinder Comprehensive Stability Analysis

The RefFinder website provides a comprehensive ranking by integrating the analysis results from various software tools to identify the most stable reference gene [28,29]. The results indicated that in the fruits at the different developmental stages, the stability ranking was as follows: *Act1* (1.19) > *UBQ* (1.86) > *GAPDH* (2.45) > *β-TUB6* (3.72) > *His3* = *UBQ10* (5.48) > *β-TUB2* (7.24) > *Act100* (8.24) > *CHS1* (8.45) > *α-TUB* (10.00). Among these, *Act1* and *UBQ* emerged as the two most stable genes, while *α-TUB* was identified as the least stable. In the distinct tissue types, the stability ranking was as follows: *Act1* (1.19) > *His3* (1.86) > *GAPDH* (2.28) > *α-TUB* (4.43) > *CHS1* (5.44) > *UBQ10* (5.83) > *β-TUB2* (6.85) > *UBQ* (7.97) > *β-TUB6* (8.35) > *Act100* (9.46). Among these, *Act1* and *His3* stood out as the two most stable genes, while *Act100* was found to be the least stable.

### 2.5. Gene Expression Analysis of the Related Enzymes in the Biosynthesis Pathway of Limonin in E. rutaecarpa var. officinalis

The candidate genes involved in the biosynthesis pathway of limonin were identified through transcriptome sequencing. When the number of candidate pathway genes exceeded two, the gene most closely associated with the limonin content was selected for investigation. In cases where there was only one candidate pathway gene, the co-correlation analysis method was not required. Limonin was not detected in the roots, leaves, or buds, and so the co-correlation analysis method could only be applied to the fruits for the different developmental stages. Taking *CYP450* as an illustrative example (Figure 8), a total of 32 *CYP450* genes were identified within the transcriptome. Subsequently, by performing co-expression cluster analyses of both the transcriptome and the metabolome data, the *CYP450* gene most likely to be involved in the limonin biosynthesis pathway was identified as *PB.10523.1*. The relative expression levels of the pathway genes in the fruits for 14 August, 1 September, and 24 October were determined, with the fruits from 17 July serving as the control. As shown in Figure 9a, the genes exhibiting decreasing expression trends encompassed *HMGCR*, *SQE*, and *CYP*, while the expression levels of the other genes displayed patterns of initial increases followed by decreases. The changing trends in the relative gene expression levels closely corresponded with the fragments per kilobase of the exon model per million mapped fragments (FPKM) values from the transcriptome data (Figure 9b).

## 3. Discussion

Research into the selection and validation of reference genes in plants is expanding. The proper selection of reference genes is a prerequisite for gene expression research. This study represents the first attempt at screening the reference genes in *E. rutaecarpa* var. *officinalis*. Among the ten commonly considered candidate reference genes in plants (*Act1*, *UBQ*, *His3*, *β-TUB6*, *GAPDH*, *UBQ10*, *Act100*, *α-TUB*, *β-TUB2*, and *CHS1*), the stability of these genes in the various tissues and fruits at the different developmental stages of *E. rutaecarpa* var. *officinalis* was comprehensively evaluated using GeNorm, NormFinder, BestKeeper, Delta Ct, and RefFinder. *UBQ* ranked second in terms of stability among the fruits at the different developmental stages, rendering it suitable as a reference gene. Its stability diminished in the different tissues, thereby making it unsuitable for reference gene usage in that context. *Act100* exhibited low stability rankings in both the fruits at the different periods and the different tissues, thereby making it unsuitable as a reference gene. Regardless of whether we were considering the fruits at the various developmental stages or the different tissues, *Act1* consistently secured the first rank in terms of stability, establishing it as the most suitable reference gene. Nevertheless, GeNorm stipulated the determination of the requisite number of reference genes when calculating the relative expression levels. Consequently, when assessing the relative expression levels in the fruits and tissues across the different development periods, the utilization of two reference genes was recommended. Many researchers have conducted relatively quantitative comparisons between the most stable and unstable reference genes, and they have often obtained significant differences. The use of unstable reference genes leads to inaccurate quantitative results [30,31,32]. Specifically, in the case of fruits at different developmental stages, the two most stable reference genes, *Act1* and *UBQ*, should be employed, while *Act1* and *His3* are advisable choices for different tissues.

The expression levels of reference genes vary among different plants and under different treatments. There is no reference gene that can be stably expressed under all conditions. For instance, in the context of the strawberry defense response, *Act1* was recommended as a suitable reference gene [33]. *Act* also performs well in cucumber for treatments involving cold and heat stress [34]. In kiwifruit, *Act1* is considered the most suitable reference gene for fruit development. However, *Act* is found to be unstable in tea plants across various experimental conditions [35], and it was ranked as the least stable reference gene in staminate and pistillate flower samples of jatropha [36]. In quinoa treated with ABA, the expression of *Act1* is also found to be unstable [37]. The identification of stable reference genes in *E. rutaecarpa* var. *officinalis* holds significant importance as it lays the foundation for the subsequent gene expression analysis.

Using the “Full length transcriptome + expression profile + metabolome” correlation analysis method for screening genes related to enzymes in biosynthesis pathways can narrow the scope of gene research. This approach is widely employed to study the synthesis mechanisms of secondary metabolites in plants. Liu et al. identified the key genes involved in flavonoid biosynthesis in the *Dendrobium* species [38]. Lou et al. identified the key genes contributing to amino acid biosynthesis in *Torreya grandis* [39]. From a pool of 80 CYP450 and 90 UDPG genes, Tang et al. screened 7 *CYP450* and 5 *UDP-glucosyltransferases (UDPG)* candidate genes that may participate in the biosynthesis of mogroside V in *Siraitia grosvenorii*. One or two of these candidates were verified through prokaryotic expression or yeast expression [40,41,42]. The FPKM values in transcriptome sequencing can reflect the expression levels of genes, and qRT PCR is used to perform secondary validations of the expression trends in pathway genes. A combination of these two methods can obtain more reliable trends in the expression levels of pathway genes. Changes in gene expression levels can cause changes in the expression levels of corresponding enzymes, which directly affect changes in secondary metabolites. During the development process of plants, there are significant changes in secondary metabolites, and therefore, gene expression patterns can be associated with plant development, and relevant pathway genes can be identified through changes in secondary metabolites.

In this study, 18 genes related to the limonin synthesis pathway were identified using this method. The expected results were that the quantitative expression levels of the pathway genes obtained through the qRT-PCR analysis would be consistent with the trends in the FPKM values in the transcriptome and that the expression levels of *HMGCR*, *SQE*, *CYP450* would be continuously decreasing, consistent with the trends in the changes in the limonin contents. The expression patterns of some genes in the MEP and MVA pathways do not completely align with the changing trends in limonin. This discrepancy may arise because these genes are simultaneously involved in the biosynthesis of evodiamine, rutaecarpin, and other compounds. The accumulation patterns of these other compounds in *E. rutaecarpa* var. *officinalis* could differ from that of limonin, or it could be due to their distant relationship with limonin synthesis and their weaker correlations. The expression levels of *HMGCR*, *SQE*, and *CYP450* were at their highest on 17 July, coinciding with the peak contents of limonin. This observation suggested that these genes were likely participants in the limonin synthesis pathway, contributing to the accelerated synthesis of limonin in plants. As the fruits ripened, the limonin contents notably decreased, accompanied by significant reductions in the expression levels of these genes. As the limonin contents reached appropriate levels, limonin biosynthesis became regulated by the reduced gene expression, leading to a gradual decline in limonin accumulation. In the quantitative results using qRT-PCR, there was an additional DXS gene with the same trend as the change in limonin content, and most of the genes had the same trend as the transcriptome, which met the expected results. The expression trends of most genes were to first increase and then decrease, which may be consistent with the metabolic law of plants. Before fruit ripening, the metabolism in fruit needs to be continuously enhanced to achieve the goal of ripening. After a fruit ripens, it begins to age, with the reduced expression levels of pathway genes and the weakened metabolism of the MVA and MEP pathways.

After screening candidate genes, it becomes essential to conduct a functional verification of these genes. From the perspectives of forward genetics and reverse genetics, these genes can be overexpressed or knocked out. Subsequently, changes in the secondary metabolite contents can be detected. Alternatively, an expression vector can be constructed and transformed into *Saccharomyces cerevisiae* or *Escherichia coli*. Inducing protein expression followed by substrate feeding can help determine whether product generation occurs. Most researchers opt for *Citrus* of the Rutaceae family as the starting point when investigating the biosynthesis mechanism of limonin. *E. rutaecarpa* var. *officinalis* belongs to the Rutaceae family as well, yet there has been limited research on the molecular mechanism of secondary metabolite biosynthesis, such as that of limonin. Investigating the molecular mechanism of limonin biosynthesis in *E. rutaecarpa* var. *officinalis* is beneficial for revealing the comprehensive limonin biosynthesis pathway.

## 4. Materials and Methods

### 4.1. Plant Materials

*E. rutaecarpa* var. *officinalis* samples were collected from the National Traditional Chinese Medicine (TCM) Production (Hunan) Technology Center of Hunan Agricultural University. The roots, leaves, flower buds, and green fruits of *E. rutaecarpa* var. *officinalis* were collected on 17 July 2019 (only a portion of the plants during this period had begun to bear fruit). The yellow–green fruits were collected on 14 August (at this stage, the fruit was still immature and belonged to the young fruit stage), the red fruits were collected on 1 September (the fruits of this period belonged to the mature period), and the purple–red fruits were collected on 24 October (the phenomenon of fruit cracking belonged to the later stage of fruit maturity). These fruits from the four different periods represented the distinct developmental stages of *E. rutaecarpa* var. *officinalis*. The plant materials were identified by Professor Liu Tasi of the Hunan University of Chinese Medicine. One portion of the materials was dried and ground into powder while the remaining portion was stored at −80 °C in a refrigerator. The powder was used to detect contents, and the portion stored in the refrigerator at −80 °C was used for RNA extraction.

### 4.2. Extraction and Relative Content Calculation of Limonin

Weigh 0.2 g of the sample powder into a conical flask, add 40 mL of 70% ethanol. Shake the mixture well and let it stand for 40 min. Proceed to extract using ultrasonics for an additional 40 min. The mixture was allowed to cool down to room temperature, and then, we collected the supernatant through a 0.45 µm filter membrane. This filtered solution was used as the test solution for further analysis. The contents of limonin were determined using high-performance liquid chromatography tandem mass spectrometry (Agilent 1290-6530Q-TOF, Agilent, Palo Alto, CA, USA). The separation of the target mixture was performed using SB C18 columns (2.1 × 100 mm, 1.8 µm; Agilent Technologies, Palo Alto, CA, USA) at a maintained temperature of 30 °C. The diode array detector (DAD) module was set to detect at 215 nm. The mobile phase A consisted of ultrapure water, and the mobile phase B was acetonitrile. The gradient elution conditions were as follows: 0~30 min, acetonitrile: 35~60%, and ultrapure water: 65~40%. The column temperature was maintained at 35 °C with a flow rate of 1 mL/min. The detection wavelength was set at 215 nm. The injection volume for both the sample solution and standard solution was 2.0 µL. The instrument was optimized in the positive electrospray ionization (ESI) mode to accurately characterize the target compound. The data were collected within the *m*/*z* range of 100~1200 Da. The gas temperature was set to 345 °C with a dry gas flow rate of 11 L/min and an atomization pressure of 50 psi, and the sheath gas temperature was set at 350 °C with a flow rate of 11 L/min, a nozzle voltage of 2.0 kV, a VCAP voltage of 4.0 kV, a fragment voltage of 175 V, an OCT1 RF Vpp of 750 V, and a drain voltage of 65 V. The MSE fragments of the typical materials were obtained using collision energy ranging from 2 to 45 eV.

### 4.3. Total RNA Extraction and cDNA Synthesis

The total RNA from the different tissues and fruits at the various developmental stages was extracted using a FastPure Universal Plant Total RNA Isolation Kit (Vazyme, Shanghai, China). The RNA integrity was evaluated through 1% agarose gel electrophoresis. The RNA concentration, A260/280 ratio, and A260/230 ratio were measured using an ultramicro spectrophotometer (BIO-DL, Shanghai, China). Subsequently, the cDNA was synthesized using a PrimeScript ™ II 1st Strand cDNA Synthesis Kit (TAKARA, Dalian, China).

### 4.4. Selection of Reference Genes and Genes Involved in Limonin Biosynthesis

The reference genes that existed in both previous reference gene screening studies and *E. rutaecarpa* var. *officinalis* were selected. These genes included *β-TUB2*, *β-TUB6*, *α-TUB*, *Act1*, *Act100*, *GAPDH*, *CHS1*, *His3*, *UBQ10*, and *UBQ*. Additionally, leveraging the “Full-length transcriptome + expression profile + metabolome” co-expression analysis, 18 genes associated with the established limonin biosynthesis pathway in *E. rutaecarpa* var. *officinalis* were identified. These genes encompassed *DXS*, *DXR*, *ISPD*, *ISPE*, *ISPF*, *ISPG*, *ISPH*, *ACAT*, *HMGCS*, *HMGCR*, *MVK*, *MVAK*, *MVD*, *IDI*, *GPS/FPS*, *SQS*, *SQE*, and *CYP450*. Primers were meticulously designed for these genes (see Table 3). All primers were designed using primer 5.0 software.

### 4.5. qRT-PCR Analysis

The qRT-PCR reactions were conducted using a Thermal Cycler Dice™ Real Time System III TP970 (TAKARA, Dalian, China), which is a real-time fluorescence quantitative PCR instrument. The total reaction volume was 25 μL, which included TB Green^®^ Premix Ex Taq™ II (Tli RNaseH Plus) (TAKARA, Dalian, China) at 12.5 μL, cDNA at 2 μL, primer at 1 μL, and RNase-free water at 8.5 μL. The preparation process was performed on ice. The cDNA template was diluted 10 times as the template before use, and the two-step PCR amplification protocol was as follows: 95 °C for 30 s (1 cycle), followed by 95 °C for 5 s and then 60 °C for 30 s (40 cycles), ending with dissociation. Each reaction was repeated three times biologically and three times technically. After the reaction, the primer’s amplification specificity was determined based on the dissociation curve analysis.

To conduct the qRT-PCR pre-experiments, cDNA from the different tissues and fruits at the various developmental stages (reverse transcribed from 1 µg RNA) were mixed. The mixed template was then diluted to concentrations of 5^−1^, 5^−2^, 5^−3^, 5^−4^, and 5^−5^ times the original concentration. These pre-experiments were carried out on the ten candidate reference genes. The standard curves were generated, yielding the slopes (*k*) and linear correlation coefficients (*R*^2^). The amplification efficiency of the reference gene primers was calculated using the formula E = (10^−1/*k*^ − 1) × 100%.

### 4.6. Analysis of the Candidate Reference Gene Expression Stability

The stability of the candidate reference genes was assessed using the following four software tools: GeNorm, Normfinder, Bestkeeper, and Delta Ct. The RefFinder website (http://blooge.cn/RefFinder/, accessed on 30 November 2022) was employed to generate a comprehensive ranking. For the relative expression analysis of the pathway genes in the fruits collected on 14 August, 1 September, and 24 October, the fruit collected on 17 July was used as the control. The quantification of the pathway gene expression levels was conducted relatively, utilizing the two most stable reference genes across the fruits at the different developmental stages (*ACT1* and *UBQ*). The 2^−ΔΔCt^ method [43] was applied to calculate the relative expression levels. The average of the two calculated values was then taken as the relative expression level of the pathway gene.

### 4.7. Stastical Analysis

The limonin biosynthesis pathway diagram was created using ChemDraw 19.0. The visualization of the histogram and line chart was performed using Graphpad Prism 8.0.2. Paired *t*-tests were utilized to calculate the significant differences. The generation of the heat map was facilitated by TBtools v1.120.

## 5. Conclusions

In this study, stable reference genes were identified in fruits at various developmental stages and in different tissues of *E. rutaecarpa* var. *officinalis*. The most stable reference genes in the fruits during the different developmental periods were *Act1* and *UBQ*, whereas in the different tissues, *Act1* and *His3* exhibited the highest levels of stability. The potential pathway genes involved in the limonin biosynthesis of *E. rutaecarpa* var. *officinalis* were determined through “Full length transcriptome + expression profile + metabolome” correlation analysis. The expression levels of these pathway genes were relatively quantified using the two most stable reference genes that were identified. A majority of the pathway genes displayed initial increases followed by decreases in expression levels. Interestingly, the genes *HMGCR*, *SQE*, and *CYP450* exhibited consistent and continuous decreases in expression levels, aligning perfectly with the trends observed in the limonin content changes. This suggested a potential role for these genes in the biosynthesis of limonin in *E. rutaecarpa* var. *officinalis*. This research is the first to investigate the molecular mechanism of effective component metabolism in *E. rutaecarpa* var. *officinalis*. Reliable reference genes were screened in *E. rutaecarpa* var. *officinalis*, and the pathway genes were quantified using the reference genes to verify the expression trends of the pathway genes. This study lays the foundation for further research on the metabolic pathway of limonin in *E. rutaecarpa* var. *officinalis*, and it provides a reliable reference gene for other researchers to use for conducting expression analyses.

## Figures and Tables

**Figure 1 plants-12-03197-f001:**
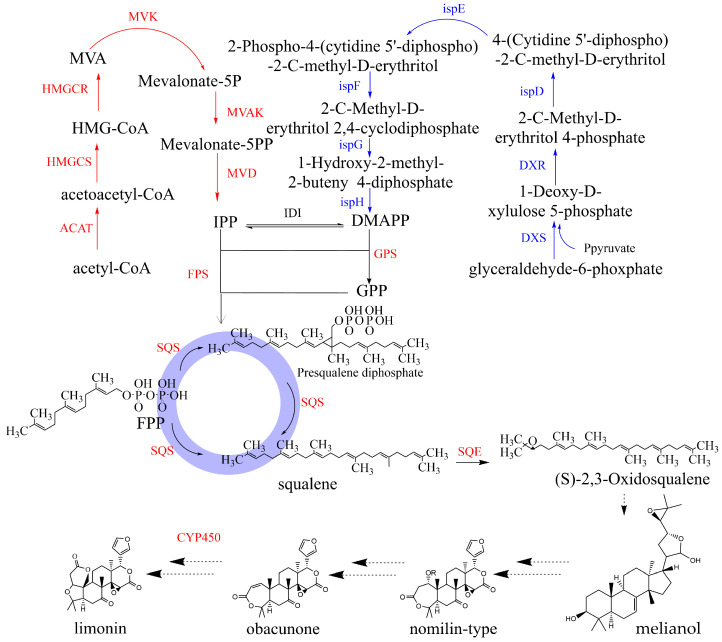
The putative biosynthesis pathway of limonin. (DXS: 1-deoxy-D-xylulose-5-phosphate synthase. DXR: 1-deoxy-D-xylulose-5-phosphate reductoisomerase. ISPD: 2-C-methyl-D-erythritol4-phosphate cytidylyltransferase. ISPE: 4-diphosphocytidyl-2-C-methyl-D-erythritol kinase. ISPF: 2-C-methyl-D-erythritol 2,4-cyclodiphosphate synthase. ISPG: (E)-4-hydroxy-3-methylbut-2-enyl-diphosphate synthase. ISPH: 4-hydroxy-3-methylbut-2-en-1-yl diphosphate reductase. FPS/GPS: Geranyl pyrophosphate synthetase/Farnesyl pyrophosphate synthetase. ACAT: acetyl-CoA C-acetyltransferase. HMGCS: hydroxymethylglutaryl-CoA synthase. HMGCR: hydroxymethylglutaryl-CoA reductase. MVK: mevalonate kinase. MVAK: phosphomevalonate kinase. MVD: diphosphomevalonate decarboxylase. IDI: isopentenyl-diphosphate Delta-isomerase. SQS: squalene synthase. SQE: squalene epoxidase. CYP450: cytochromeP450). The full sequences of all the genes are available in the Appendix A.

**Figure 2 plants-12-03197-f002:**
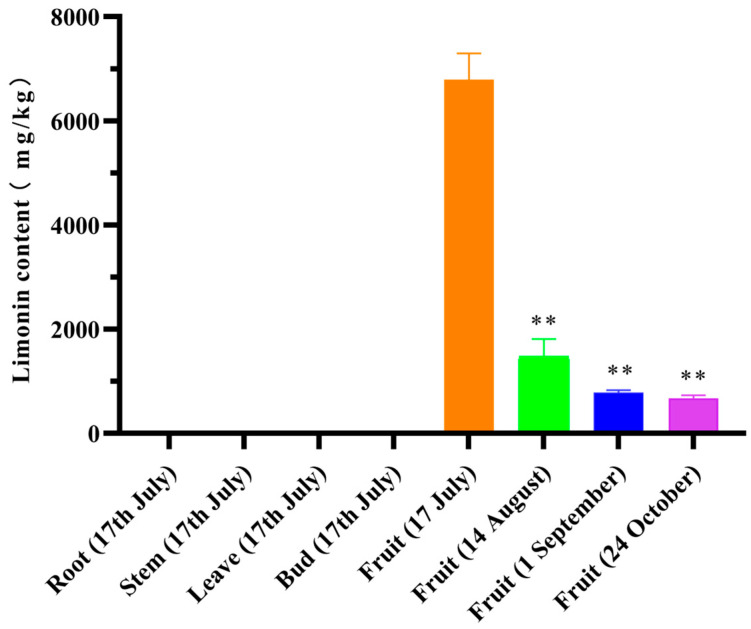
Relative contents of limonin in the different tissues and fruits for the different periods (** indicates *p* < 0.01).

**Figure 3 plants-12-03197-f003:**
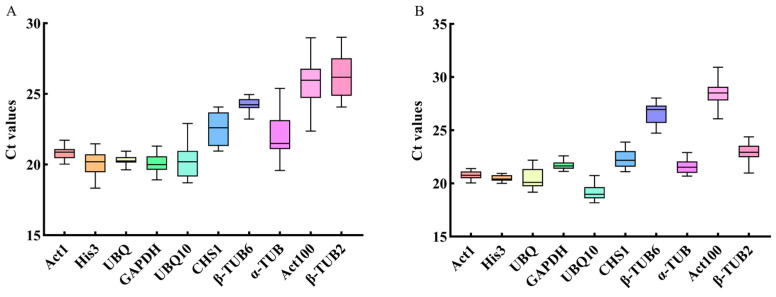
Ct value distribution box-plots of the ten candidate reference genes in the fruits for the four periods (**A**) and four tissue types (**B**).

**Figure 4 plants-12-03197-f004:**
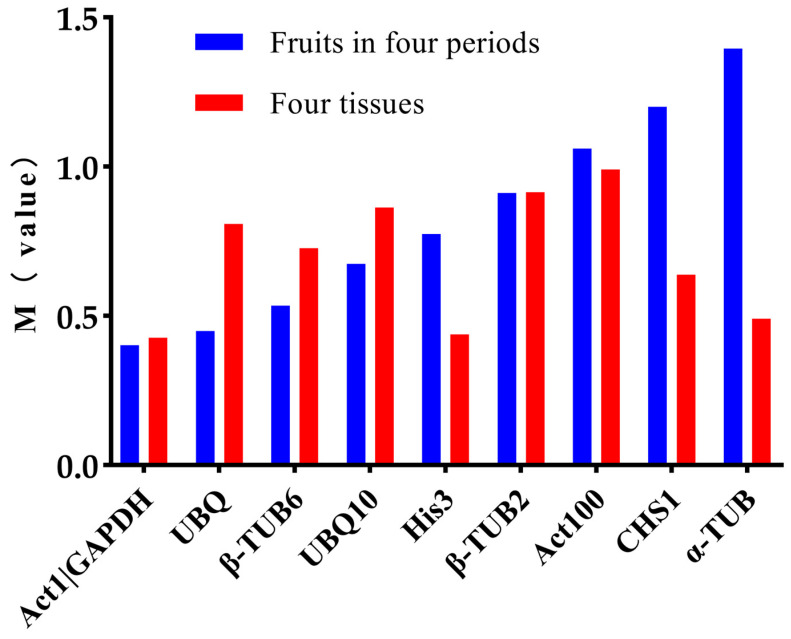
The stability analysis results of the ten reference genes using the GeNorm software. (M value represents the calculated stable value; usually, when the M value is less than 1.5, the gene can be considered a stable reference gene).

**Figure 5 plants-12-03197-f005:**
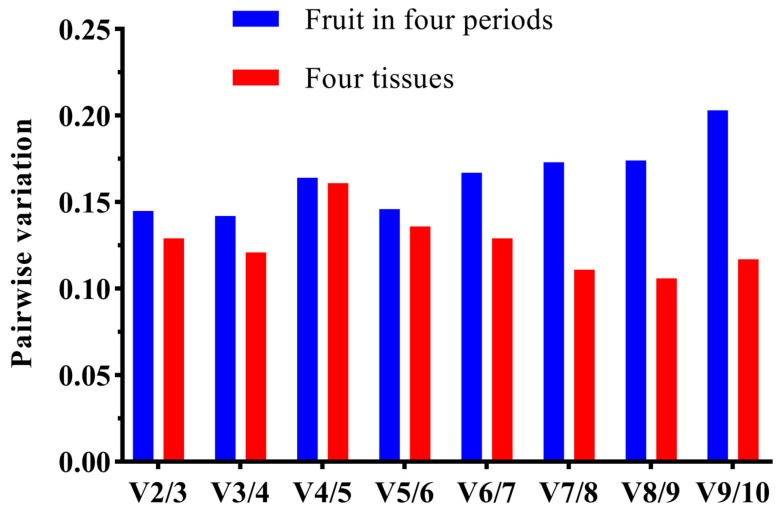
Determination of the optimal number of control genes for normalization. (When *Vn*/*n*+1 is less than 0.15, *n* internal reference genes are required as quantitative standards).

**Figure 6 plants-12-03197-f006:**
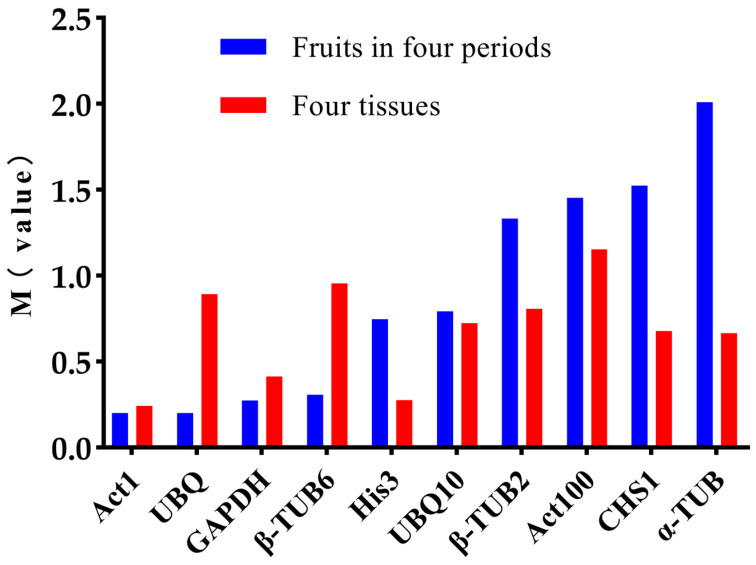
NormFinder stability analysis of the ten candidate reference genes. (Normfinder is similar to GeNorm in that it also calculates a stable value M, where a smaller M value indicates that the reference gene is more stable).

**Figure 7 plants-12-03197-f007:**
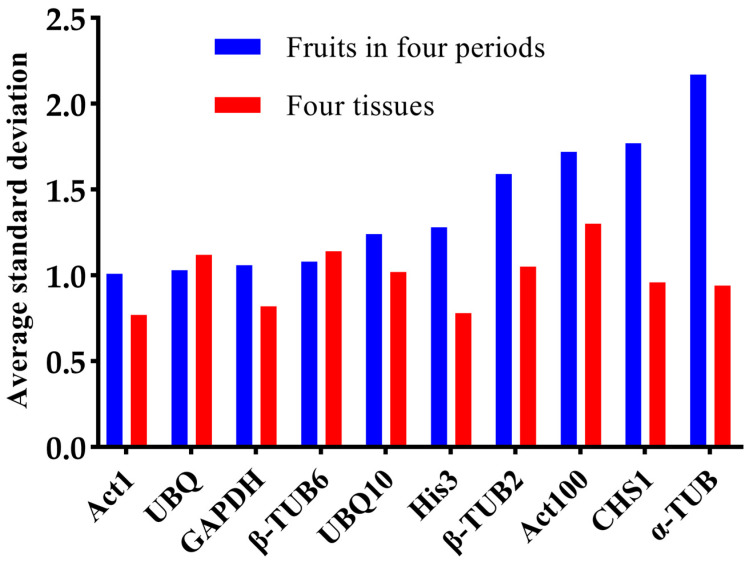
Delta Ct stability analysis of the ten candidate reference genes. (DeltaCt can directly use the Ct value to calculate the average standard deviation to evaluate the stability of genes, and the smaller the average standard deviation of the reference gene, the more stable it is).

**Figure 8 plants-12-03197-f008:**
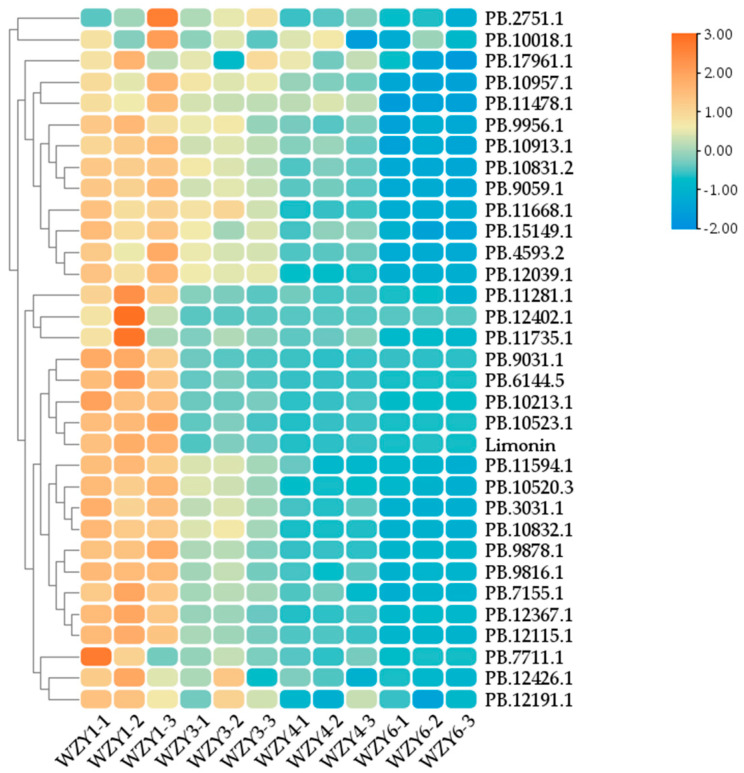
Screening of cytochrome P450 associated with limonin biosynthesis. (WZY1−1, WZY1−2, and WZY1−3 represent the three repeats of limonin content in the *E. rutaecarpa* var. *officinalis* fruit on 17 July. WZY3−1, WZY3−2, and WZY3−3 represent the three repeats of limonin content in the *E. rutaecarpa* var. *officinalis* fruit on 14 August. WZY4−1, WZY4−2, and WZY4−3 represent the three repeats of limonin content in the *E. rutaecarpa* var. *officinalis* fruit on 1 September. WZY6−1, WZY6−2, and WZY6−3 represent the three repeats of limonin content in the *E. rutaecarpa* var. *officinalis* fruit on 24 October).

**Figure 9 plants-12-03197-f009:**
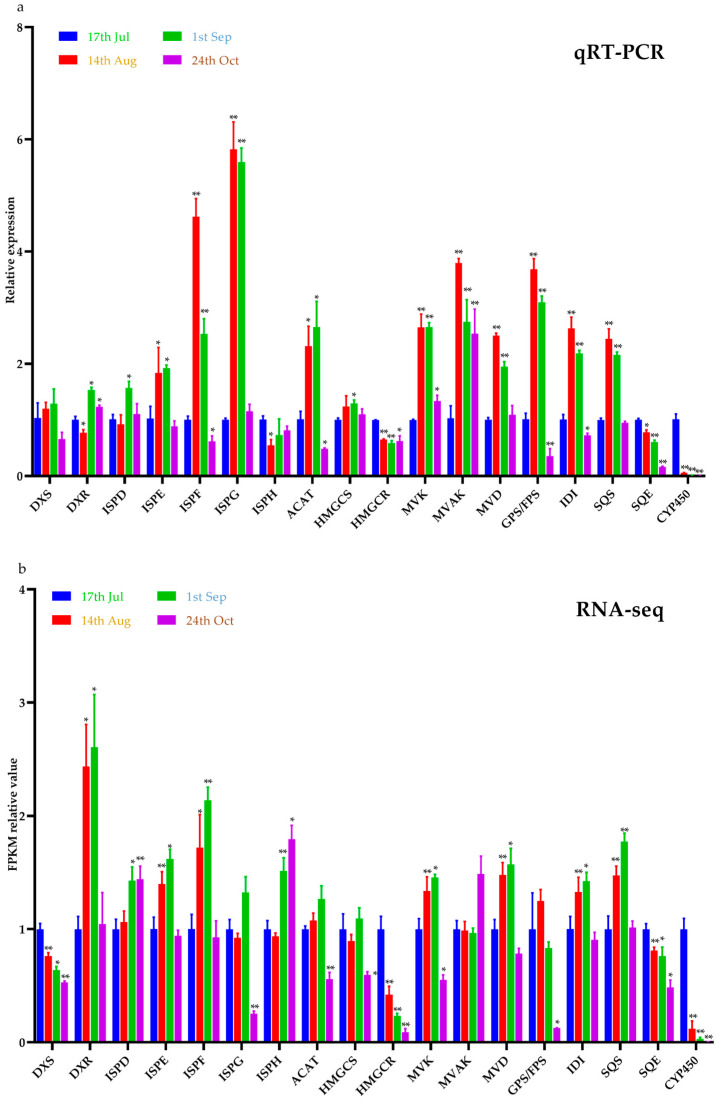
Analysis of the expression patterns of the genes related to the limonin synthesis pathway. (**a**) The expression pattern calculated by the qRT-PCR. (**b**) The expression pattern calculated by the FPKM values in the transcriptome (** indicates *p* < 0.01 and * indicates *p* < 0.05).

**Table 1 plants-12-03197-t001:** Primer sequence, correlation coefficients, and amplification efficiency of qRT-PCR for the ten candidate reference genes of *E. rutaecarpa* var. *officinalis*.

Gene	Full Name	Primer Sequence (5′ to 3′)	Slope (*k*)	Amplification Efficiency (%)	Correlation Coefficients (*R*^2^)
*α-TUB*	α-tubulin	F: TCCTTTCAACCCATTCCC R: CTCGGTCACATCCACATTCA	−3.157	107.4	0.996
*β-TUB2*	β-tubulin2	F: GCTACCTCACAGCCACAG R: CCATTTCATCCATACCCT	−3.113	109.5	0.990
*β-TUB6*	β-tubulin6	F: AGTCGTGGAGCCATACAA R: AACGGAATCAGATTAACAGC	−3.140	108.2	0.996
*His 3*	Histone H3	F: TAAATCAACTGGAGGAAAGG R: GAGACCGACGAGGTAAGC	−3.074	111.5	0.990
*CHS 1*	Chalcone synthase 1	F: GCAAAGAAGCAGCCACCA R: TTAACGGAAGGGCGAAGG	−3.176	106.5	0.998
*UBQ10*	Ubiquitin10	F: CCAGAAAGAATCCACGCTAC R: CACGGAGACGAGAACAAGA	−3.128	108.8	0.998
*UBQ*	Ubiquitin	F: AAACCCTAACTGGGAAGAC R: TACCACGTAGACGAAGCAC	−3.335	99.5	0.997
*Act1*	Actin1	F: TCCGTGACATGAAGGAGA R: AGAAATGGCTGGAAGAGG	−3.248	103.2	0.991
*Act100*	Actin100	F: TTTCCCTTTATGCCAGTG R: TTTCCCGCTCAGCAGTAG	−3.465	94.4	0.985
*GAPDH*	Glyceraldehyde-3-phosphate dehydrogenase	F: AACTGATTGGTATGGCTTTC R: CTCGGACTCCTCCTTGAT	−3.193	105.7	0.997

**Table 2 plants-12-03197-t002:** Expression stability of the ten candidate reference genes using BestKeeper (coefficient of variation (CV) ± standard deviation (SD)).

Rank	Fruits in Different Periods		Different Tissues	
	Gene	CV ± SD	Gene	CV ± SD
1	*UBQ*	1.16 ± 0.24	*His3*	1.16 ± 0.24
2	*Act1*	1.7 ± 0.35	*Act1*	1.55 ± 0.32
3	*β-TUB6*	1.49 ± 0.36	*GAPDH*	1.55 ± 0.34
4	*GAPDH*	2.62 ± 0.53	*UBQ10*	2.68 ± 0.51
5	*His3*	3.43 ± 0.69	*β-TUB2*	2.34 ± 0.54
6	*UBQ10*	4.44 ± 0.9	*α-TUB*	2.57 ± 0.55
7	*CHS1*	5.3 ± 1.19	*CHS1*	3.08 ± 0.69
8	*β-TUB2*	4.74 ± 1.24	*Act100*	2.76 ± 0.79
9	*Act100*	4.91 ± 1.26	*UBQ*	3.92 ± 0.8
10	*α-TUB*	6.12 ± 1.35	*β-TUB6*	3.43 ± 0.91

**Table 3 plants-12-03197-t003:** qRT-PCR primer sequences related to limonin biosynthesis pathway.

Gene	Full Name and the E.C. Number	Primer Sequence (5′ → 3′)
*DXS*	1-deoxy-D-xylulose-5-phosphate synthase (EC: 2.2.1.7)	F:ACTGCCACCAGAAAACAAA
		R:GCTACGAACGAGGGAATGA
*DXR*	1-deoxy-D-xylulose-5-phosphate reductoisomerase (EC: 1.1.1.267)	F:TCCATCCGTGAATCTTTG
		R:GCTGCGTAATCTCGTGCC
*ISPD*	2-C-methyl-D-erythritol4-phosphate cytidylyltransferase (EC: 2.7.7.60)	F:AGGATTTCCATACCCAAGA
		R:CAATCGGTTGACCCAGAA
*ISPE*	4-diphosphocytidyl-2-C-methyl-D-erythritol kinase (EC: 2.7.1.148)	F:CCACCAACTCCATCTACCT
		R:CCTGTCTACTTCATCTGCC
*ISPF*	2-C-methyl-D-erythritol 2,4-cyclodiphosphate synthase (EC: 4.6.1.12)	F:CGTCCAAATCAAAATCACT
		R:AATCGCATCCACAACACAG
*ISPG*	(E)-4-hydroxy-3-methylbut-2-enyl-diphosphate synthase (EC: 1.17.7.1)	F:CAGAAGCACCAGAGGAGGA
		R:CCAGAACGACGATGAAAAT
*ISPH*	4-hydroxy-3-methylbut-2-en-1-yl diphosphate reductase (EC: 1.17.7.4)	F:AAATCATTCACAACCCGAC
		R:CCAGACCTTAGACACCCAA
*FPS/GPS*	Geranyl pyrophosphate synthetase/Farnesyl pyrophosphate synthetase (EC: 2.5.1.1 2.5.1.10)	F:GGAAAAAATATACGAGGCA
		R:AGACAGGGCAAGTAGAGCA
*ACAT*	acetyl-CoA C-acetyltransferase (EC: 2.3.1.9)	F:CCTGGTGGAAGAGGGAAAC
		R:GAGCCGCATCAGCAAATC
*HMGCS*	hydroxymethylglutaryl-CoA synthase (EC: 2.3.3.10)	F:TGGACATCTACTTCCCTCG
		R:CACAGTTTCGCTTCCTACT
*HMGCR*	hydroxymethylglutaryl-CoA reductase (EC: 1.1.1.34)	F:CGACGAGAACAAAGAGAAG
		R:GTAAAACGCAACGGAAAAG
*MVK*	mevalonate kinase (EC: 2.7.1.36)	F:ACACGAAAGTTGGGAGAA
		R:TATGGAGGAATGGCTGAC
*MVAK*	phosphomevalonate kinase (EC: 2.7.4.2)	F:TCAGAAGTAGCAGAGCCAT
		R:GTATCAAGAACACGAGCAT
*MVD*	diphosphomevalonate decarboxylase (EC:4.1.1.33)	F:TCGTAGTCTTTTTGGTGGA
		R:TGTATTATGCGTTTCGGTA
*IDI*	isopentenyl-diphosphate Delta-isomerase (EC: 5.3.3.2)	F:CAGCCATCCTCTATTCCGC
		R:ATCAAGTTCATGCTCCCCC
*SQS*	squalene synthase (EC: 2.5.1.21)	F:TGACACCAGCATACCTAC
		R:AATCGCCTCCTGATAACC
*SQE*	squalene epoxidase (EC: 1.14.14.17)	F:GGTTCGGTGTCTGGTTGA
		R:GCTTCTATTTGGCATTGTTT
*CYP450*	cytochromeP450 (1.14.x.x)	F:GGTCAGGTAGACGGGTTT
		R:TCTCATCCGACGGTAGCC

## Data Availability

Not applicable.

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
