# Peer review of "Selection of Reference Genes in Evodia rutaecarpa var. officinalis and Expression Patterns of Genes Involved in Its Limonin Biosynthesis"

_plants, 2023, doi:10.3390/plants12183197_

Round 1

Reviewer 1 Report

The authors investigated the genes involved in limonin biosynthesis using molecular biology tools. The manuscript is well-written however has several main issues, including the quality of the figures and some missing information, which are listed below:

(1) Concerning the developmental stages, the dates 17th July, 22nd August, 1st September, and 24th October provide inadequate information.

(2) Please assess the scientific accuracy of the sentence: "Terpenoids are polymers."

(3) The sentence "All isoprenoids are derived from the common precursor isopentenyl diphosphate (IPP)" is incomplete, as it does not mention the role of dimethylallyl pyrophosphate.

(4) The font size in Figure 1 is very small, making it difficult to read the figure.

(5) The text "early stage (17th July), middle stage (14th August), late stage (1st September), and final stage (24th October)" is inconsistent with the x-axis of Figure 2.

(6) Many abbreviations are not defined the first time they appear (e.g., Ct and others).

(7) For the construction of Figure 4, a dashed line is more suitable due to the absence of a mathematical model to define the data's behavior. Considering the absence of a continuous variable on the x-axis, a bar graph may be more appropriate.

(8) Regarding limonin quantification, it is unclear why the results are not expressed in mass/mass units (relative area units were used instead of concentration units). An analytical standard of limonin is available.

(9) In Figure 9, it is unclear which groups are being compared by a statistical test (** indicates p < 0.01, * indicates p < 0.05). The data analysis methodology described only includes a t-test as a mean comparison test; however, I believe that more than two groups are being compared in Figure 9. Please verify this and simplify the figure's interpretation. The same comment may be addressed to the Figure 2.

(10) The captions of Figures 4-7 need better explanations regarding their methodology and interpretation. This will enhance readability and understanding for readers.

(11) Please add the E.C. number of the enzymes investigated in Table 3.

Minor editing of English language required

Author Response

Thank you for your affirmation of this work and for your suggestions on the revisions to the manuscript. The details of the revisions are attached.Please see the attachment.

Reviewer 2 Report

Comments

In this study, the authors analyzed the biosynthesis pathway of limonin, the main active component of the plant E. rutaecarpa var. officinalis. For this purpose, is first assessed the stability of ten candidate reference genes. These genes were identified through a screening process involving "full-length transcriptome + expression profile + metabolome" analysis. After that, the expression profile of the genes involved in the limonin biosynthesis pathway was established. Based on the observed changes in gene expression, a conclusion was made about the potential participation of several genes in limonin biosynthesis in E. rutaecarpa var. officinalis.

The work is meaningful and well written, but there are some shortcomings that need to be addressed.

-        The content of limonin was determined using high performance liquid chromatography. I suggest that the authors present the obtained results of relative content of limonin in different tissues and fruit in different periods in Table by mg/kg.

-        Figure 1 is extremely unclear, prepare new Figure 1 with increase the font size for the text on the image.

-        Section 4.5 describes the Quantitative Real-Time PCR (qRT-PCR) experiment. It is more correct that the title of this part should be qRT-PCR analysis or experiment than qRT-PCR conditions.

-        The sentence in line 150 contains a full stop instead of a comma.

-        Check the space between the text and the brackets in the name of the ordinate in Figures 2, 4, and 6.

English language is fine, minor correction are needed.

Author Response

(The authors gave the same response as above.)

Reviewer 3 Report

Reviewer Reports:

I recommend a major amendment at this level.

General comments:

The manuscript entitled “Selection of reference genes in Evodia rutaecarpa var.officinalis  and expression patterns of genes involved in its limonin bio-synthesis” was reviewed. The work carried out in the manuscript is interesting and aimed at assessing the stability of ten candidate reference genes and utilizing the two most stable ones as benchmarks for the relative quantification of genes associated with the limonin biosynthesis pathway. Better connect your research findings to previous works published in Plants and in other top journals. It is better to do not to use the first-person's pronoun. Do not use "we, us, or our" throughout the paper. The innovation and the importance of this work are not clearly highlighted in the abstract, introduction, and conclusions. Please work on this and prove to us why this work is valuable. Please also remove ANY lumped references. Please define each of them separately to avoid inappropriate citations. It is recommended that the authors work with a science editor who is proficient in the Native English language to improve the organization and delivery of some portions of the manuscript. The journal's author guidelines and instructions should be followed in preparing the revised version. Other main remarks that in my opinion needs attention are the following:

Detailed comments:

The abstract should state briefly the purpose of the research, the principal results, and major conclusions. In the abstract, please add an indication of the achievements from your study that are relevant to the journal's scope. Please be concise - maximum 1-2 lines. The abstract should include a sentence about your findings, discussions, and conclusions in your abstract and underscore the scientific value added to your paper in your abstract.

The review of the literature needs more updating with works to have a clear and concise state-of-the-art analysis. This should more clearly show the knowledge gaps identified and link them to the paper's goals. The introduction section is poorly organized. While the general introduction is acceptable, the state-of-the-art review that follows is very difficult to understand and no specific thoughts can be inferred. The major defect of this study is the debate or argument is not clearly stated. Could you explain the rationale behind choosing Evodia rutaecarpa var. officinalis as the subject of this study? What makes it a suitable candidate for investigating limonin biosynthesis? You may see these articles and follow them in the revised version. The relevant reference may be of interest to the author according to below: https://doi.org/10.1016/j.envres.2023.115919; https://doi.org/10.1007/s11356-022-24176-1; Please eliminate the use of redundant words. Eg. In this way, Recently, Respectively, therefore, currently, thus, hence, finally, to do this, first, in order, however, moreover, nowadays, today, consequently, in addition, additionally, furthermore. Please revise all similar cases, as removing these term(s) would not significantly affect the meaning of the sentence. This will keep the manuscript as CONCISE as possible. Please check ALL. Avoid beginning or ending a sentence with one or a few words, they are usually redundant. Kindly revise all. How were the candidate reference genes identified and validated for normalization of gene expression data in Evodia rutaecarpa var. officinalis? What specific criteria were used for their selection?

Please avoid having one heading after another with no discussion in between as in the case of Sections 4 and 4.1. Kindly inspect the entire document for similar instances and revise accordingly. Please add in the beginning of your scientific hypothesis. In the course of describing the performed actions, please provide reader guidance, sufficient for understanding why those actions have been performed.  Could you elaborate on the methodology employed to assess the expression patterns of genes involved in limonin biosynthesis? How were these patterns correlated with different stages of plant development or environmental conditions?

The structure of this work should be reorganized. For example, the Section of results should be combined with the Discussion. The authors are suggested to have the results and discussion part together. All the findings of the current work need to be compared and discussed with the results of other researchers finding instead of having a general comparison with other researchers' works. The authors should perform a comparison between the forecasting results. In your discussion section, please link your empirical results with a broader and deeper literature review. Were there any unexpected or intriguing findings regarding the expression patterns of the genes involved in limonin biosynthesis? How do these findings contribute to our understanding of the metabolic pathways in Evodia rutaecarpa var. officinalis?

Please make sure your conclusions section underscores the scientific value-added of your paper, and/or the applicability of your findings/results. Highlights the novelty of your study. In the conclusions, in addition to summarising the actions taken and results, please strengthen the explanation of their significance. It is recommended to use quantitative reasoning compared with appropriate benchmarks, especially those stemming from previous work. In the context of gene expression analysis, what challenges can arise from using inadequate reference genes? How might these challenges impact the interpretation of results in this study?

Please check the reference section carefully and correct the inconsistency. Please update this section.

Need moderate revision. 

Author Response

(The authors gave the same response as above.)

Round 2

Reviewer 3 Report

Reviewer 1:

I have reviewed the revised version manuscript entitled “Selection of reference genes in Evodia rutaecarpa var.officinalis and expression patterns of genes involved in its limonin biosynthesis”. The paper has been improved and can be accepted.

The new version was refined and this current version ok in this format.